# The Periodontopathic Pathogen, *Porphyromonas gingivalis*, Involves a Gut Inflammatory Response and Exacerbates Inflammatory Bowel Disease

**DOI:** 10.3390/pathogens11010084

**Published:** 2022-01-11

**Authors:** Yu-Chen Lee, Chih-Yi Liu, Chia-Long Lee, Ruo-Han Zhang, Chi-Jung Huang, Ting-Lin Yen

**Affiliations:** 1Department of Dentistry, Cathay General Hospital, Taipei 106, Taiwan; Cgh12673@cgh.org.tw; 2Division of Pathology, Sijhih Cathay General Hospital, New Taipei City 221, Taiwan; cyliu@cgh.org.tw; 3School of Medicine, College of Medicine, Fu Jen Catholic University, New Taipei City 242, Taiwan; 4Department of Internal Medicine, Cathay General Hospital, Taipei 106, Taiwan; cghleecl@cgh.org.tw; 5Graduate School of Health Industry Management, Ching Kuo Institute of Management and Health, Keelung City 20301, Taiwan; 1101609012@ms.cku.edu.tw; 6Department of Oral Hygiene Care, Ching Kuo Institute of Management and Health, Keelung City 20301, Taiwan; 7Department of Medical Research, Cathay General Hospital, Taipei 106, Taiwan; 8Department of Biochemistry, National Defense Medical Center, Taipei 114, Taiwan; 9Department of Pharmacology, School of Medicine, College of Medicine, Taipei Medical University, Taipei 110, Taiwan

**Keywords:** periodontal disease, *Porphyromonas gingivalis*, inflammatory bowel disease, gut inflammation

## Abstract

Periodontal disease (PD) is one of the most prevalent disorders globally and is strongly associated with many other diseases. Inflammatory bowel disease (IBD), an inflammatory condition of the colon and the small intestine, is reported to be associated with PD through undetermined mechanisms. We analyzed taxonomic assignment files from the Crohn’s Disease Viral and Microbial Metagenome Project (PRJEB3206). The abundance of *Porphyromonadaceae* in fecal samples was significantly different between patients with Crohn’s disease and control volunteers. Dextran sulfate sodium was used to induce colitis in mice to reveal the effect of this periodontopathic pathogen in vivo. After intrarectal implantation of *Porphyromonas gingivalis* (*Pg*)—the primary pathogen causing PD—the disease activity index score, colonic epithelial loss, and inflammatory cell infiltration were intensified. In addition, tumor necrosis factor-α and interleukin-6 showed the highest levels in *Pg-*infected colons. This revealed the importance of *Pg* in the exacerbation of IBD. Thus, simultaneous treatment of PD should be considered for people with IBD. Moreover, implantation of *Pg* in the rectum worsened the clinical symptoms of colitis in mice. Because *Pg* participates in the pathogenesis of IBD, reducing the chances of it entering the intestine might prevent the worsening of this disorder.

## 1. Introduction

Periodontal disease (PD) involves the development of lesions on the periodontium (gingiva, alveolar bone, cementum, and periodontal ligament) [1]. This oral disease involves a wide range of inflammatory conditions caused by bacterial biofilm [2]. The severity of PD increases with age, and affected individuals typically have other comorbidities [2,3,4]. Thus, improving the prevention and control of PD involves the core governance activities of public health assessment, policy development, and insurance [5].

Epidemiologic studies have shown a positive association between PD and overall cancer risk [6]. For example, PD might worsen tumor-promoting effects and has been linked to many human cancers, such as oral, esophageal, lung, prostate, and liver cancers [7,8,9,10]. The increased cancer risk may be associated with PD by microbial stimulation of a destructive inflammatory response [3,11,12].

Inflammatory bowel disease (IBD), which includes Crohn’s disease (CD) and ulcerative colitis (UC), is correlated with an increased risk of colorectal cancer [13]. These chronic, relapsing–remitting systemic diseases cause similar clinical burdens and have similar treatment strategies [14]. Anti-inflammatory agents can protect against IBD [15]. On the other hand, PD and IBD are often found to have a comorbid relationship, and modulating excessive inflammation [6,16,17,18,19] might have potential in PD treatment [20]. Because there is dysbiosis of the microbiota in these two inflammatory diseases [17,19,21], microbiota might help explain the link between PD and IBD. Such imbalance in host–microbiota interactions might lead to undesired immune responses to the intestinal microbiota and result in chronic bowel inflammation [22,23,24]. The bacteria present in individuals with gingivitis can penetrate deeper into the tissues and surrounding periodontium. This triggers a host immune response against the invading bacteria [1].

*Porphyromonas gingivalis* (*Pg*) is known to be a pathogen that contributes to the initiation and progression of PD [25,26]. Besides PD, several systemic pathologies, such as cardiovascular diseases, rheumatoid arthritis, pancreatic cancer, nonalcoholic steatohepatitis, and neurodegenerative pathologies, have been linked to *Pg* infection [27,28,29,30]. In this regard, IBD and PD are both chronic inflammatory diseases [31]. However, the underlying pathological and molecular network between IBD and PD is not fully understood. Therefore, understanding the influence of *Pg* in IBD might help patients with PD to evade IBD or prevent its deterioration. Here, we hypothesized that *Pg* might play a role in the clinical signs of IBD, and a dextran sulfate sodium (DSS)-induced mouse model of colitis was applied to test the possible causal relationship between *Pg* infection and IBD.

## 2. Results

### 2.1. Porphyromonadaceae Showed Higher Abundance in the Feces of Patients with Crohn’s Disease

We downloaded taxonomic assignment files from the Crohn’s Disease Viral and Microbial Metagenome Project (PRJEB3206), which comprise 11 amplicon-sequenced files from fecal samples of patients with CD and eight amplicon-sequenced files from control fecal samples of volunteers. The extracted information included the bacterial taxonomic composition and abundance at the family level. At the end of the analysis, we compared samples from the two groups. The abundance of Porphyromonadaceae in fecal samples was significantly different between the patients with CD and control volunteers. In the feces of patients, the abundance of Porphyromonadaceae (Figure 1) was substantially higher than in the control samples. This observation suggested that *Pg*, a member of this family, might influence the clinical signs of CD.

### 2.2. Effect of Pg on Clinical Signs of Colitis Induced by DSS

It is well-established that CD is a major type of IBD [32]. The DAI is a score that grades weight changes, stool consistency, and rectal bleeding to assess the clinical evolution of colitis. As shown in Figure 2, after four days of 4% DSS administration, there was no statistically significant difference between all groups. After seven days of 4% DSS treatment, the *Pg*-treatment groups showed a slight increase in the DAI score but no significant difference with the control (no treatment) group. The 4% DSS treatment group demonstrated the clinical sign of IBD for having two points on the DAI score. Compared with the 4% DSS treatment group, 3 points of DAI score was recorded in the 4% DSS plus *Pg* group, a significant difference.

Colon length has been proven to be a biological marker in assessing colonic inflammation: the shorter the colon, the more severe the colitis [33]. Figure 3A illustrates the colons of each group; the length of colons was measured at the end of the experiment (seven days after treating DSS). In Figure 3B, there was no significant difference between the control and *Pg*-alone groups; the 4% DSS treatment group had shorter colons, and the shortest colon was found in the 4% DSS plus *Pg* group. DSS treatment clearly caused colitis, but *Pg* infection by itself showed no marked effect on clinical signs of colitis. However, in the 4% DSS plus *Pg* group, the highest DAI score, and the shortest colon revealed that *Pg* treatment exacerbated the inflammatory response of the colon and was associated with IBD. The data in Figure 4 demonstrated that the clinical signs, including colon length and DAI score, showed similar patterns.

### 2.3. Pg Implantation Exacerbated DSS-Mediated Colitis by IHC Score

Epithelial changes, overall mucosal architecture, and the degree of inflammatory cell infiltration are the symptoms of intestinal inflammation in mouse models [34]. Here we evaluated the surface epithelial loss, crypt destruction, and inflammatory cell infiltration into the mucosa (Figure 4A) and summed the score of these parameters as the mucosal damage score (Figure 4B). H&E staining was used to demonstrate the histopathology pattern of the colon under different treatments.

The mice with active colitis induced by 4% DSS treatment showed mucosal erosion with crypt architectural distortion and mixed infiltrates of acute and chronic inflammatory cells. *Pg* implantation, together with 4% DSS treatment, resulted in severe active inflammation with pronounced epithelial loss, marked crypt architectural alterations, and dense cellular infiltrates. Significant inflammatory cell infiltrates and crypt architectural changes were absent in both the control and *Pg*-alone groups, but the IHC scores were not significantly different. The score of the 4% DSS treatment group was statistically significantly different from the control group. Moreover, the 4% DSS plus *Pg* group had the highest score, which was significantly different from the treatment group receiving 4% DSS alone. Thus, *Pg* implantation without 4% DSS treatment did not affect the integrity of colonic tissues. However, *Pg* treatment worsened mucosal damage when colitis was induced by 4% DSS treatment.

### 2.4. Overexpression of Tumor Necrosis Factor Alfa (TNF-α) and Interleukin (IL-6) in Pg-Infected Colons with Colitis

The levels of TNF-α and IL-6 are positively correlated with disease severity in patients with IBD [35,36]. TNF-α causes villous atrophy, decreases epithelial cell turnover, and increases cell apoptosis [37]. In addition, neutralizing IL-6 might effectively treat IBD [38]. For understanding the effect of *Pg* in colitis, we examined the expression level of each group of colons in this study. The results shown in Figure 5A demonstrated abundant inflammatory cells immunostained for TNF-α in the 4% DSS plus *Pg* group, with a higher expression level than in the 4% DSS treated group. Figure 5B shows the staining intensity of IL-6, with a marked increase in the 4% DSS plus *Pg* group, more than in the 4% DSS treated group. Compared with the 4% DSS plus *Pg* group, only a few IL-6-positive cells were found in the *Pg*-alone group.

## 3. Discussion

PD is one of the most prevalent conditions worldwide, and it is reported to be related to IBD, although the mechanism of their relationship is still undetermined. Some studies have elucidated the relationship between PD and UC. Here, we found that *Pg*, the primary pathogen of PD, might serve to link PD and IBD. The presence of *Pg* increased the DAI score, colon epithelial loss, and inflammatory cell infiltration during DSS-induced colitis. In addition, the highest expression levels of TNF-α or IL-6 in mouse colons cotreated with DSS and *Pg* suggested that *Pg* intensified the expressions of TNF-α and IL-6.

Successful PD treatment is associated with a significantly reduced overall risk of cancer and reduced risks of certain types of cancers [39]. In previous studies, antibiotic treatments were found to delay the progression of rheumatoid arthritis and Alzheimer’s Disease in patients with PD [40,41]. Moreover, several systematic reviews and meta-analyses of clinical trials demonstrated that antibiotics such as ciprofloxacin and metronidazole [42] appear to effectively induce remission in patients with IBD [43,44]. Therefore, the use of antibiotics for inhibiting the pathogens causing PD, including minocycline, doxycycline, and metronidazole [45], might be considered for treating patients with IBD under an appropriate dosage to avoid gastrointestinal side effects.

In this study, *Pg* treatment by itself did not affect the integrity of the mouse colon or the expression of the proinflammatory cytokines TNF-α and IL-6. However, it showed an adverse effect in colons with mucosal damage caused by DSS. According to this result, without protecting the intestinal mucosal barrier, *Pg* showed more pathogenicity in the gut. Thus, the diagnosis of PD is vital for the prognosis and treatment of patients with IBD. However, the *Pg*-derived lipopolysaccharide (LPS) surprisingly showed partial benefit in DSS-induced murine colitis. In the previous studies, *Pg* LPS downregulated the expression of *Lgr5*, a representative marker for intestinal stem cells, and *Alpi*, a marker for absorptive enterocytes to modulate the differentiation of epithelial cells [46]. This finding showed another view of *Pg* in colitis, making researchers reconsider the role of *Pg* in colitis. In the future, more detailed mechanisms of *Pg* LPS should be investigated.

Some studies demonstrated that PD was most common in the elderly, tobacco smokers, and populations from high-income countries [47,48]. Therefore, the treatment of PD must be considered when IBD occurs in these populations.

Although animal models do not perfectly recapitulate human IBD, they have led to the discovery of important concepts in its pathogenesis [49]. Much recent progress in the understanding of immunity has been achieved using experimental animal models of intestinal inflammation [49]. Experimental colitis can be induced using chemical irritants such as DSS or transgenic mouse models. The picture of all the intestines in each group are presented in the Appendix A. Essentially, DSS does not cause intestinal inflammation directly; instead, it injures the intestinal epithelium, resulting in exposure of the lamina propria and submucosal compartment to luminal antigens and enteric bacteria, triggering inflammation [33,50,51] Thus, UC can be induced in mice by varying the concentration and duration of DSS administration in drinking water [33,52].

The limitation of this work was that there were not enough animals in this study. According to the principle of 3R (replacement, reduction, refinement), we inspected the fewest animals and obtained accurate data. To observe the diverse effects made by the target microbe, such as the major player in the current study, the *Porphyromonas gingivalis*, further enlarged sample size examination will be conducted in the future.

In conclusion, *Pg* plays a critical role in the progression of colitis. *Pg* implantation in the rectum induced overexpression of TNF-α and IL-6, followed by loss of surface epithelium, destruction of crypts, and infiltration of inflammatory cells, thereby exacerbating the clinical symptoms of colitis in this mouse model. Given the prevalence of PD, clinicians treating individuals susceptible to IBD should consider PD treatment simultaneously. Furthermore, since *Pg* participates in the pathogenesis of IBD, the importance of using antibiotics for treating PD, such as minocycline and metronidazole, needs to be further explored in the future.

## 4. Materials and Methods

### 4.1. Data Collection

We obtained the bacterial 16S rRNA sequencing data from the Crohn’s Disease Viral and Microbial Metagenome Project (PRJEB3206). The project information is available from the EMBL database (https://www.ebi.ac.uk/metagenomics/studies/MGYS00000307) (accessed on 30 August 2021). Vicente et al. documented the use of 16S rRNA gene sequences for bacterial identification, taxonomic analysis, and estimating diversity, and their approach was followed here [53].

### 4.2. Mouse Model of DSS-Induced Colitis

Male BALB/c mice (6–8 weeks old) were purchased from the National Laboratory Animal Center (Taipei, Taiwan) and maintained in the Animal Research Center, according to the Institutional Animal Care and Use Committees (approval no. IACUC 110-011) at Cathay General Hospital (Taipei, Taiwan). All animals were housed in plastic cages (two or three per cage) in controlled conditions of humidity (50 ± 10%), light (12/12 h light/dark cycle), and temperature (23 ± 2 °C). The mice were quarantined for seven days before being randomized by body weight into different groups for testing *Pg*: (1) subgroup with no treatment (control group, *n* = 3); (2) subgroup with *Pg* (1 × 10^8^ colony forming units (CFU) in 100 μL) administration through anal injection (*Pg* alone group, *n* = 3); (3) subgroup with acute DSS-induced UC by drinking water containing 4% DSS (molecular weight, 36,000–50,000; MP Biomedicals, Solon, OH, USA) for seven days (4% DSS group, *n* = 3); (4) subgroup with *Pg* administration as subgroup 2 and 4% DSS administration as subgroup 3 (4% DSS + *Pg* group, *n* = 3). Mice were finally euthanized by high dose halogenated anesthetics (isoflurane: delivered via vaporizer for anesthetic induction (3%, 5 min) and euthanasia (5%, 15 min)). The colons were collected after confirmation of cessation of respiratory. All efforts were made to minimize the number of animals and their suffering.

### 4.3. Inflammation in Tissues of Mice with DSS-Induced Colitis

Mice with induced colitis were euthanized, and the abdominal cavity was opened. The colon was isolated and opened longitudinally. The inflammation status was characterized according to macroscopic damage, histopathology, and inflammation-related cytokines. Briefly, macroscopic damage to the entire colon sections was assessed from the degrees of body weight loss, stool consistency (diarrhea), and colonic bleeding using a previously established scoring system (disease activity index, DAI) [54]. Then, histopathology and immunohistochemical (IHC) staining were performed. Isolated colons were fixed, dehydrated, and embedded in paraffin wax using standard techniques, and 5-μm thick sections were transferred to slides. Histology was performed on the sections after hematoxylin and eosin (H&E) staining. The histological features of (i) surface epithelial loss, (ii) crypt destruction, and (iii) inflammatory cell infiltration into the mucosa were used to determine the severity of mucosal damage. According to the widely accepted scoring system, the damage was scored from 0 (normal) to 4 (extensive and severe) [55].

### 4.4. Immunohistochemical Staining and Semi-Quantitative Determination of Protein Expression

For IHC staining to detect various proteins, TNF-α and IL-6 were respectively immune-detected with anti-TNF-α antibody (1:100 in phosphate-buffered saline; Cat No. GTX15821; GeneTex, Irvine, CA, USA) and anti-IL-6 antibody (1:100 in phosphate-buffered saline; Cat No. ab208113; Abcam, Cambridge, UK) for 32 min at 37 °C under an automated Ventana Benchmark XT (Ventana Medical Systems, Tucson, AZ, USA). Labeling was detected with the Optiview DAB Detection Kit (Ventana Medical Systems, Inc., Tucson, AZ, USA) according to the manufacturer’s instruction. All sections were counterstained with hematoxylin in Ventana reagent (Ventana Medical Systems). After being dehydrated and mounted, the imaging results were scored by histopathologists.

The staining frequency of proteins was semiquantitatively scored based on the percentages of positive cells. The scoring criteria were as follows: (i) TNF-α, 0–10% of cells stained, score 0; 11–25% of cells stained, score 1; 26–50% of cells stained, score 2; 51–100% of cells stained, score 3 [56]; (ii) IL-6, 0% of cells stained, score 0; 1–25% of cells stained, score 1; 26–50% of cells stained, score 2; 51–100% of cells stained, score 3 [57].

### 4.5. Pg Preparation

We commissioned the Bioresource Collection and Research Center (Hsinchu, Taiwan) to culture *Pg* following their protocols. Here we applied the most widely used chemical colitogen, DSS, to induce colitis in mice. To measure the effect of *Pg* on colitis, *Pg* (1 × 10^8^ CFU) was administered intrarectally to mice with DSS-induced colitis, and the changes in disease activity index (DAI) scores were measured. As shown in the scheme (Figure 6), days one and three before DSS treatment, mice in *Pg* insulted groups (2 and 4) were administered *Pg* (1 × 10^8^ CFU) by anal injection. Subsequently, mice in groups 3 and 4 were treated with 4% DSS for the next seven days. *Pg* insulted groups (2 and 4) were continued to receive *Pg* (1 × 10^8^ CFU) by intrarectal injection at days one, three, and five after DSS treatment.

### 4.6. Statistics

The significance of any differences between the two groups was assessed using Student’s *t-test*. The Kruskal–Wallis one-way analysis of variance by ranks test determined whether four independent groups were significantly different. The commercial statistical package, SPSS (ver. 22.0, IBM SPSS statistics), was used in these analyses. Data are presented as means ± standard errors of the mean for the indicated number of independently performed experiments, and *p* < 0.05 was considered statistically significant.

## Figures and Tables

**Figure 1 pathogens-11-00084-f001:**
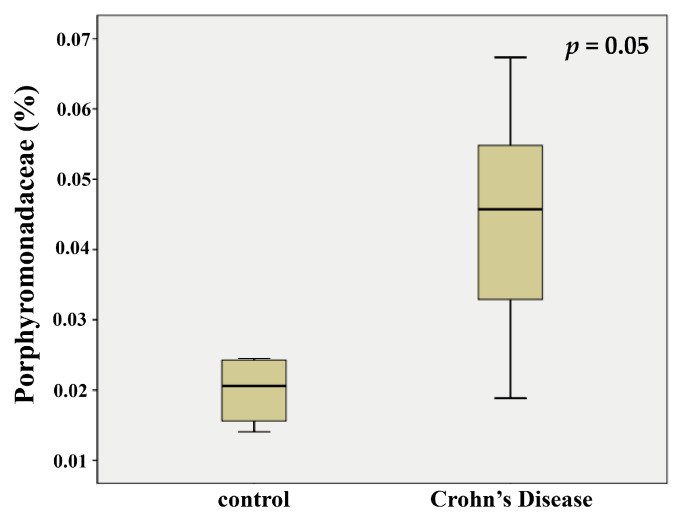
The abundance of *Porphyromonadaceae* in patients with or without Crohn’s disease. Bacterial 16S rRNA sequencing data was obtained from the Crohn’s Disease Viral and Microbial Metagenome Project (PRJEB3206). The abundance of *Porphyromonadaceae* in fecal samples showed significant differences between the patients with CD (*n* = 11) and control volunteers (*n* = 8). Data were presented as the means ± SD. *p* = 0.05, compared with the control (volunteers) group.

**Figure 2 pathogens-11-00084-f002:**
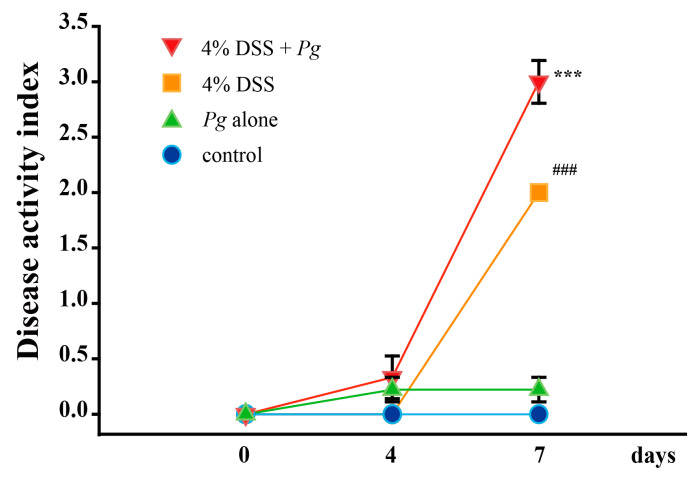
Analyzing the disease activity index (DAI) change after 4 and 7 days of treatment with 4% DSS. 4% DSS insulted animals showed no statistically significant difference in DAI between all groups at day 4. After seven days of 4% DSS treatment, the *Pg*-treatment alone group (group 2) showed in the DAI score a slight increase but no significant difference with the control (no treatment, group 1) group. The 4% DSS treatment group (group 3) had two points, and 4% DSS plus *Pg* group (group 4) had three points in the DAI score. Data were presented as the means ± SD (*n* = 3). ^###^
*p* < 0.001, compared with the control group; *** *p* < 0.001, compared with the 4% DSS group.

**Figure 3 pathogens-11-00084-f003:**
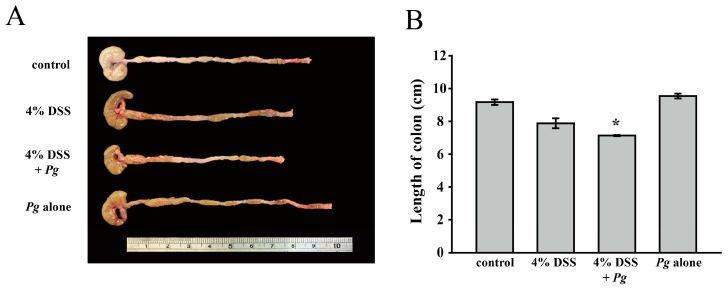
Examine the effect of *Pg* and DSS on the variety of colon lengths. (**A**) Photograph of dissected colon (with cecum) specimen was a representative example of three similar experiments. (**B**) The colon lengths of each group were presented respectively as the means ± SD of the means (*n* = 3). * *p* < 0.05, compared with the 4% DSS group.

**Figure 4 pathogens-11-00084-f004:**
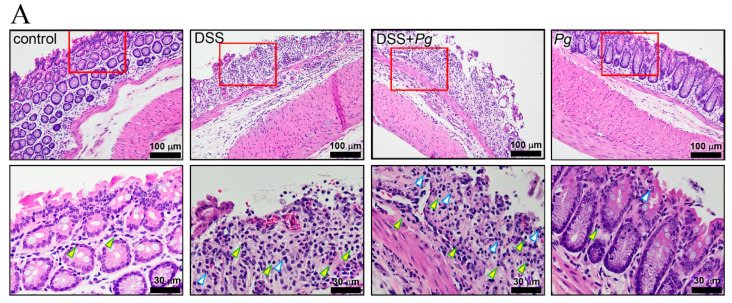
Histologic examination of colon tissue samples. (**A**) Representative photomicrographs of colon specimens were representative examples of three similar experiments. The yellow arrow indicated the mononuclear leukocytes, and the polymorphonuclear leukocytes were indicated by the white arrow. (**B**) The mucosal damage score of each group was calculated and presented respectively as the means ± SD of the means (*n* = 3), ^#^ *p* < 0.05, compared with the control group; * *p* < 0.05, compared with the 4% DSS group.

**Figure 5 pathogens-11-00084-f005:**
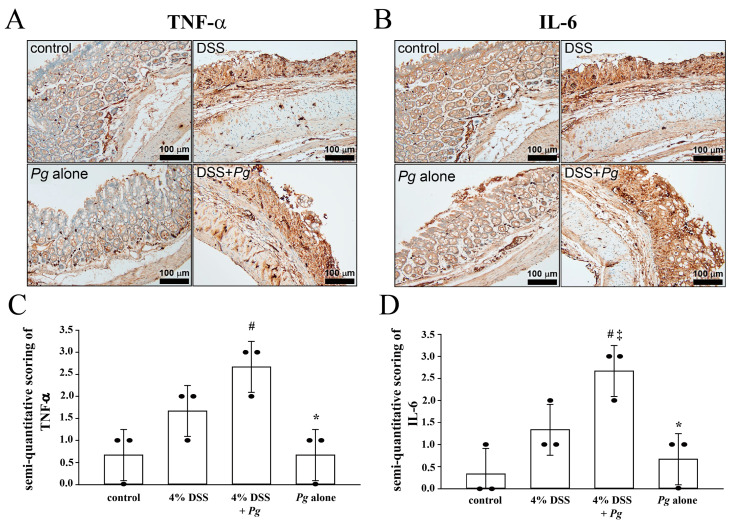
Expression pattern of TNF-α and IL-6 proteins. Immunohistochemistry (IHC) analysis stated the presence and location of TNF-α and IL-6 proteins in colon specimens. (**A**) TNF-α was stained remarkably in groups administrated with DSS, and DSS + *Pg* insulted colon showed the highest expression of TNF-α. (**B**) IHC labeled IL-6 protein in DSS *+ Pg* insulted colon showed higher intensity level than the colon treated with DSS alone. Semi-quantitative scoring of (**C**) TNF-α and (**D**) IL-6 proteins. The score of each group was calculated and presented respectively as the means ± SD of the means (*n* = 3): ^#^ *p* < 0.05, compared with the control group in (**C**,**D**); * *p* < 0.05, compared with the 4% DSS plus *Pg* group in (**C**,**D**); ^‡^ *p* < 0.05, compared with the 4% DSS group in (**D**).

**Figure 6 pathogens-11-00084-f006:**
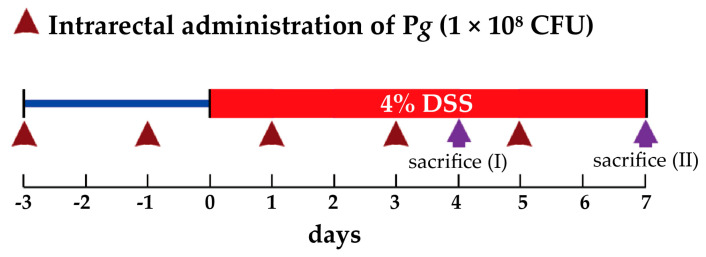
Schematic of the treatment protocols. Animals in groups 3 and 4 were treated with 4% DSS (dissolved in drinking water) from day 0 to day 7. For transplanting the *Pg*, animals in groups 2 and 4 were administrated *Pg* (1 × 10^8^ CFU in 100 μL) by the intrarectal injection method. *Pg* was preadministrated three days before day 0 (initiation of 4% DSS treatment) and followed every two days until the end of the examination. Animals were sacrificed (euthanasia) on day 4 for DAI score assessment and day 7 for colon length determination and histological analysis.

## Data Availability

The datasets used and analyzed during the current study are available from the corresponding author on reasonable request.

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
