# Peer review of "The Periodontopathic Pathogen, *Porphyromonas gingivalis*, Involves a Gut Inflammatory Response and Exacerbates Inflammatory Bowel Disease"

_pathogens, 2022, doi:10.3390/pathogens11010084_

Round 1

Reviewer 1 Report

This manuscript reports that anal challenge with P. gingivalis exacerbates chemically-induced colitis in a murine model of inflammatory bowel disease. This is consistent with the observation that the bacterial family Porphyromonadaceae, which includes the genus Porphyromonas, is enriched in the fecal flora of IBD patients. This is a very small observational study (one-off experiment, 3 mice per group), so the results are not very robust. There are several areas (noted in specific comments) where the authors overclaim the interpretation of their data. There is little attempt to approach the potential role of P. gingivalis in IBD from a mechanistic standpoint, or for that matter, to suggest how a microbe that, even in periodontal disease, constitutes a very small component of the oral microflora would have a significant effect in the colon.

Specific comments:

  1. Lines 52-53: There is a major difference between causation and association. While in some cases there appears to be an association between cancer risk and periodontal disease, the cited studies do not establish that any increased cancer risk associated with periodontal disease is caused by microbial stimulation of a destructive inflammatory response.
  2. Lines 70-71: The cited reference reported studies in mice. It reported no data on detection of gingivalis in the gut of “affected patients.” Please remove this highly misleading statement.
  3. Discussion: Many investigators hypothesize that gingivalis LPS is responsible for much of its pro-inflammatory activity. While the present study suggests that P. gingivalis may exacerbate IBD, the authors should discuss the study by Seo et al. (doi.org/10.1186/s42826-019-0029-6) that P. gingivalis LPS had some protective effects against DSS-induced murine colitis.
  4. Lines 185-186: The present study did not report the presence of gingivalis in the stools of patients with IBD. Please remove this highly misleading statement.
  5. Lines 186-188: The authors might temper their enthusiasm for prophylactic antibiotic treatment for IBD given that reference #45 also clearly describes the well-known gastrointestinal side effects of several of the suggested antibiotics.

Author Response

Dear reviewer:
Please see the attachment.

Reviewer 2 Report

Interesting study

Author Response

(The authors gave the same response as above.)

Reviewer 3 Report

The authors focused on the association of PD with IBD and investigated the effect of Pg, one of the causative agents of PD, on IBD using a DSS-induced mouse model. The experiments were conducted to prove the association between PD and IBD, and suggesting that treatment of PD can contribute to the improvement and prevention of exacerbation of IBD. I think the idea of ​​research and the way of advancing experiments are appropriate. However, the description of the number of subjects and the information on the results are a little inadequate, so I think this paper needs to be revised.

Major points

1. Please describe the approval number for ethical statement in animal experiments in Materials and methods.

  1. Could you show the picture of all the intestines in each group in Figure 3? Alternatively, increase the number of mice in the experimental group. I think it can judge the result more objectively.

  1. In Figure 4, please show me a photo with a slightly higher magnification, and show some inflammatory cells with arrows.

  1. The authors are observing TNF-a and IL-6 in the tissue, but are they actually quantifying those cytokines secreted in the intestinal tract by ELISA or the like?

Minor points

  1. Please indicate the sample size of each group in the footnote of Figure 1.

  1. Please explain in a little more detail how to express the mucosal damage score (Reference # 53) in materials and methods.

Author Response

(The authors gave the same response as above.)

Round 2

Reviewer 1 Report

The authors have adequately addressed the reviewers' concerns.

Author Response

Dear reviewer:
Please see the attachment.

Ting-Lin Yen

Reviewer 3 Report

Thanks for the authors' response.

My further opinions are as follows.

  1. Please show the picture of all the intestines in each group as a supplementary figure.
  2. Please indicate the results of the significance test in Figure 5 C and D as shown in Figure 4.

Author Response

(The authors gave the same response as above.)
